# Integrative analysis of the metabolome and transcriptome provides insights into the mechanisms of amino acids and their derivatives biosynthesis in *Polygonatum*

**Xiuzhi Wang, Lingjun Cui, Qiang Xiao** [ID] *

Hubei Provincial Key Laboratory of Occurrence and Intervention of Rheumatic Diseases (Hubei Minzu University), Enshi, China

* 1992022@hbmzu.edu.cn

## Abstract

The composition and concentration of amino acids and their derivatives in *Polygonatum* are crucial factors that indicate its nutritional value and have a substantial impact on its potential for market development as a medicinal and edible plant. This study utilized comprehensive targeted metabolomics detection, transcriptome analysis, and weighted gene co-expression network analysis (WGCNA) to assess the quality characteristics of amino acids and their derivatives. Additionally, it aimed to investigate the pivotal genes that influence the synthesis of these compounds. The analysis revealed that there was a total of 72 distinct amino acids and their derivatives present in both forms of *Polygonatum*. Furthermore, there were notable variations in the content of 37 of these amino acids. A total of 271 genes are responsible for encoding 37 enzymes that are involved in the production of amino acids and their derivatives. The WGCNA clustering analysis, along with the metabolic profile of amino acids and their derivatives and transcriptome analysis, categorized the transcripts into six modules. Among these modules, the active components of amino acids and their derivatives showed a positive correlation with the MEturquoise and MEblue modules. This correlation helped identify 10 key genes that are highly likely to influence the synthesis of amino acids and their derivatives. Therefore, we validated these results using qRT-PCR. The modulation of the expression patterns of these crucial genes can establish a theoretical foundation for the subsequent breeding of *Polygonatum* germplasm with abundant amino acid content and diversity.

## Introduction

*Polygonatum kingianum var.* grandifolium (PzP) and *Polygonatum sibiricum* Redouté (PS) are both members of the Liliaceae family, specifically belonging to the genus *Polygonatum* [1]. This is a valuable traditional Chinese medicinal substance that is

**Data availability statement:** All relevant data are within the paper and its Supporting Information files.

**Funding:** This study was jointly funded by the National Natural Science Foundation of China [NSFC, grant number 31260057, QX), Natural Science Foundation of Hubei Province (Joint Fund) [NSFHP, grant number 2023AFD077, QX], The Open Fund of Hubei Key Laboratory of Biological Resources Protection and Utilization (Hubei Minzu University) [OHBP, grant number KYPT012403, LJC]. The funders had no role in study design, data collection and analysis, decision to publish, or preparation of the manuscript.

**Competing interests:** The authors have declared that no competing interests exist.

extensively utilized in the realm of traditional Chinese medicine. The flavor of this substance is characterized by a combination of sweetness, bitterness, and a mild warmth. It is believed to have an effect on the spleen, lung, and kidney meridians. the ability to restore energy and nourish the body's vital essence, as well as promote the health of the bones and bone marrow [2]. *Polygonatum* has a diverse range of compounds such as flavonoids [3], saponins, alkaloids [4], and polysaccharides [5], which exhibit notable pharmacological effects. Furthermore, the scientific community has shown significant interest and conducted extensive research on the abundant amino acids and their derivatives found in *Polygonatum* [6].

*Polygonatum* rhizomes contain a diverse range of amino acids and their derivatives. These chemicals play a crucial role in protein synthesis in the body and also have many physiological activities and pharmacological effects that are important for human health. Research has demonstrated that serine and alanine are highly prevalent free amino acids and their derivatives in *Polygonatum*. Additionally, *Polygonatum* multiflorum contains amino acids and their derivatives such as aspartic acid, homoserine, and diaminobutyric acid [7,8]. At present, our knowledge about how the genes control the production of these amino acids and their derivatives in *Polygonatum* is still restricted. Research has shown that amino acids and their derivatives are one of the initial substances produced by living organisms [9]. This could be the reason why they play a role in many metabolic pathways and cellular activities [10]. In addition, amino acids and their derivatives are used as building blocks for the production of other secondary metabolites, enzymes, and hormones [11]. Research has found that the production of amino acids and their derivatives relies on chemicals derived from carbohydrate metabolism. Additionally, the breakdown of amino acids and their derivatives results in the release of several metabolites that serve as energy sources in the citric acid cycle [12]. Moreover, amino acids and their derivatives have a substantial impact on pharmacology. Research has demonstrated that asparagine (Asn), glutamine (Gln), arginine (Arg), methionine (Met), serine, and cysteine play significant roles in the metabolic processes of cancer cells. They function as targets for anticancer therapies, by binding to disease sites in order to achieve therapeutic effects [13]. Further investigation reveals that the breakdown of amino acids and their byproducts is linked to immunological responses, obesity (specifically insulin resistance), and the regulation of thermogenesis [14]. Prior studies on traditional Chinese medicinal materials have primarily concentrated on investigating the medicinal properties of the herbs, specifically examining the mechanisms of action of polysaccharides, flavonoids, saponins, alkaloids, and other molecules. However, these studies have overlooked the significance of amino acids and their derivatives, which also exhibit numerous biological functions.

This work utilized metabolite profiling and transcriptomics analysis to examine the crucial genes and regulatory impacts in the biosynthesis pathways of amino acids and their derivatives in *Polygonatum*. Through the utilization of WGCNA clustering and correlation analysis, we have successfully discovered crucial genes that may play a role in the production of amino acids and their derivatives. This research provides novel insights into the regulatory mechanisms of amino acid and derivative production in *Polygonatum* species.

## Materials and methods

### Plant materials

In this study, we used three-year-old *Polygonatum* plants, specifically PS and PzP, which were grown under similar conditions at the experimental base of the College of Forestry and Horticulture, Hubei Minzu University. The substrate soil was composed of vermiculite and peat soil (in a ratio of 8:1). Three biological replicates of different *Polygonatum* rhizomes were collected for each group. The sampled *Polygonatum* rhizomes were chopped, mixed evenly, placed into sterile centrifuge tubes, quickly frozen in liquid nitrogen, and then stored at −80°C until use. They were subsequently used for RNA sequencing and metabolite profiling analysis.

### Metabolite extraction and analysis

The data acquisition instrument system mainly comprises an ultra-performance liquid chromatography (UPLC) system (SHIMADZU Nexera X2) and a tandem mass spectrometer (MS/MS) (Applied Biosystems 4500 QTRAP).

Liquid Chromatography Conditions: (i) Column: Agilent SB-C18 1.8 μm, 2.1 mm×100 mm; (ii) Mobile Phase: Phase A is ultra-pure water (with 0.1% formic acid added), Phase B is acetonitrile (with 0.1% formic acid added); (iii) At 0.00 min, the proportion of Phase B is 5%, linearly increased to 95% within 9.00 min, and maintained at 95% for 1 min; from 10.00 to 11.10 min, the proportion of Phase B is decreased to 5%, and equilibrated at 5% until 14 min; (iv) Flow rate: 0.35 mL/min; column temperature: 40°C; injection volume: 4 μL. The UPLC-MS/MS was conducted by Metware Biotechnology Co., Ltd. (Wuhan, China). Metabolomics data were acquired in electrospray ionization negative (ESI-) and positive (ESI+) modes. The ion spray voltage for ESI- is −4500 V, and for ESI+ it is 5500 V; the ion source gases I (GSI), II (GSII), and curtain gas (CUR) are set to 50, 60, and 25 psi, respectively, with collision-induced dissociation parameters set to high; and the electrospray ion source (ESI) temperature is 550°C. Additionally, principal component analysis (PCA), partial least squares-discriminant analysis (PLS-DA), orthogonal partial least squares-discriminant analysis (OPLS-DA), differential metabolite expression analysis, and Kyoto Encyclopedia of Genes and Genomes (KEGG) pathway analysis were performed on the metabolomics data.

### Transcriptome analysis

Total RNA was extracted from the rhizomes of *Polygonatum* using the Plant RNA Kit (200)R6827-02 (OMEGA, USA). A cDNA library was constructed and then sequenced on the Illumina platform. After obtaining clean reads, Trinity was used to assemble the clean reads. Following the assembly, the longest cluster sequence obtained by corset hierarchical clustering was used as unigene for subsequent analysis. The Unigene sequences were aligned with the KEGG, NCBI non-redundant protein sequences (Nr), a manually annotated and reviewed protein sequence database (Swiss-Prot), Gene Ontology (GO), Clusters of Orthologous Groups of Proteins (COG)/euKaryotic Ortholog Groups (KOG), a variety of new documentation files, and the creation of TrEMBL (TrEMBL) databases using the DIAMOND BLASTX software to predict the amino acid sequences of the Unigenes. After the prediction, the HMMER software was used to align with the Protein family (Pfam) database to obtain annotation information for the Unigenes. The gene expression levels were assessed using fragments per kilobase of transcript per million fragments mapped (FPKM), and differentially expressed genes (DEGs) were defined.

### Transcriptome and metabolome association analysis

To identify key genes associated with the synthesis of amino acids and their derivatives, we conducted an integrated analysis of differentially expressed genes (DEGs) and the differential accumulation of amino acids and their derivatives in PzP and PS. Pearson correlation coefficients were calculated between the differential metabolites and DEGs. The co-expression network was visualized using Cytoscape (version 3.9.1).

## Quantitative Real-Time PCR (qRT–PCR) validation

To verify the accuracy of the transcriptome data, We selected the sequences of 10 target genes identified in the study for primer design and performed qRT-PCR analysis. We extracted RNA using ComWin Biotech (Beijing, China) Plant All-in-One RNA Extraction Kit, and reverse transcribed RNA into cDNA using RTIII All-in-One Mix with dsDNase Reverse Transcription Kit from Monad Biotech (Wuhan, China). primers were synthesized by Sangon Biotech (Sangon, Shanghai, China). The experiment was conducted using an ABI7500 real-time fluorescence quantitative PCR apparatus. The real-time quantitative PCR reaction system consisted of 20 μL, comprising 1 μL of cDNA, 8 μL of RNase-free water, 10 μL of 2X SGExcel FastSYBR Master, and 0.5 μL of each forward and reverse primers. The reaction program involved a prede-naturation step at 95°C for 3 minutes, followed by 40 cycles of denaturation at 95°C for 5 seconds and annealing at 60°C for 20 seconds. The annealing/extension step should be performed at a temperature of 60°C for a duration of 20 seconds. The *ubiquitin* was selected as a reference gene: forward primer: 5'- GGACCCAGAAGTACGCAATG-3', reverse primer: 3'- AATTACCAGGGATACAGCACC-5' [15]. Three technical replicates were prepared for each extract, The complete list of primers may be seen in S1 Table.

## Results and analysis

### Metabolomic analysis of PzP vs PS

We conducted an analysis of metabolite alterations using UPLC-MS/MS by selecting two distinct samples. A comprehensive targeted metabolomics approach identified a total of 72 amino acid and derivative metabolites in the metabolic analysis (S2 Table). Fig 1A and 1B provide a summary of the analysis of metabolites and the principal component analysis (PCA) of amino acids and their derivatives comparing PzP and PS. The clustering heatmap demonstrates that the six *Polygonatum* samples can be categorized into two distinct groups (Fig 1A). Significant disparities in metabolite content exist between PzP and PS, underscoring the major variations in composition between the two kinds. In the PCA score plot, the first two principal components (PC1 at 87.69% and PC2 at 4.36%) have a combined contribution of 92.05% (Fig 1B). The plots clearly demonstrate distinct clustering of the PzP and PS types, emphasizing the considerable metabolic disparities between them. The tight grouping of duplicate samples for each variation highlights the experiment's repeatability and its suitability for subsequent qualitative and quantitative examination.

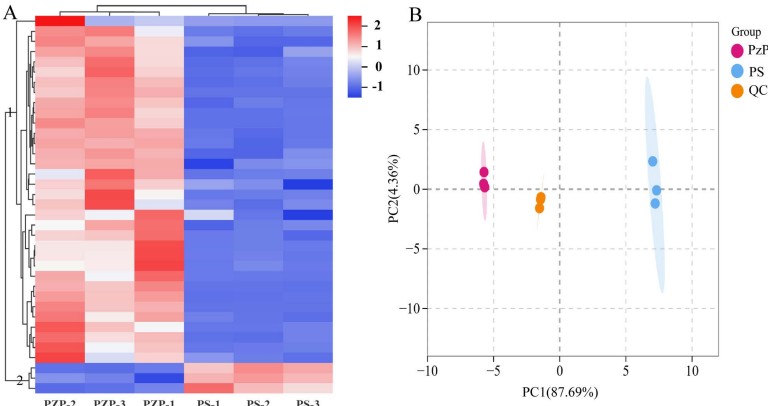

**Fig 1. Multivariate statistical analyses of DEMs.** A: Hierarchical cluster analysis was performed on the identified amino acids and their derivative metabolites in the metabolomes of the three samples. The colors represent the accumulation levels of each metabolite, ranging from low (blue) to high (red). The Z-score indicates the deviation in standard deviation units from the mean. B: Principal component analysis (PCA) of the identified amino acids and their derivative metabolites.

## Detection of amino acids and their derivatives that show differential accumulation in PzP vs PS

To reveal the changes in metabolite abundance among different *Polygonatum*, a quantitative analysis and inter-group comparison were performed on all identified metabolites. The metabolomics data were analyzed using the OPLS-DA model to construct the score plots for each group, further highlighting the differences between groups (Fig 2A). The values of R2X, R2Y, and Q2 being close to 1 indicate that the model is stable and reliable. To identify differential metabolites, we selected metabolites from the OPLS-DA model with a fold change (FC) ≥ 2 (upregulated) or ≤ 0.5 (downregulated) and a variable importance in the projection VIP value ≥ 1. CV is a measure of data dispersion. As shown in Fig 2B, more than 80% of the QC samples have a CV value below 0.2, indicating high stability of the experimental data. Compared with the PS rhizomes, statistical analysis identified 37 differentially accumulated metabolites (DAMs) of amino acids and their derivatives in PzP, including 34 upregulated and 3 downregulated (Fig 2C; S3 Table). Interestingly, most amino acids and

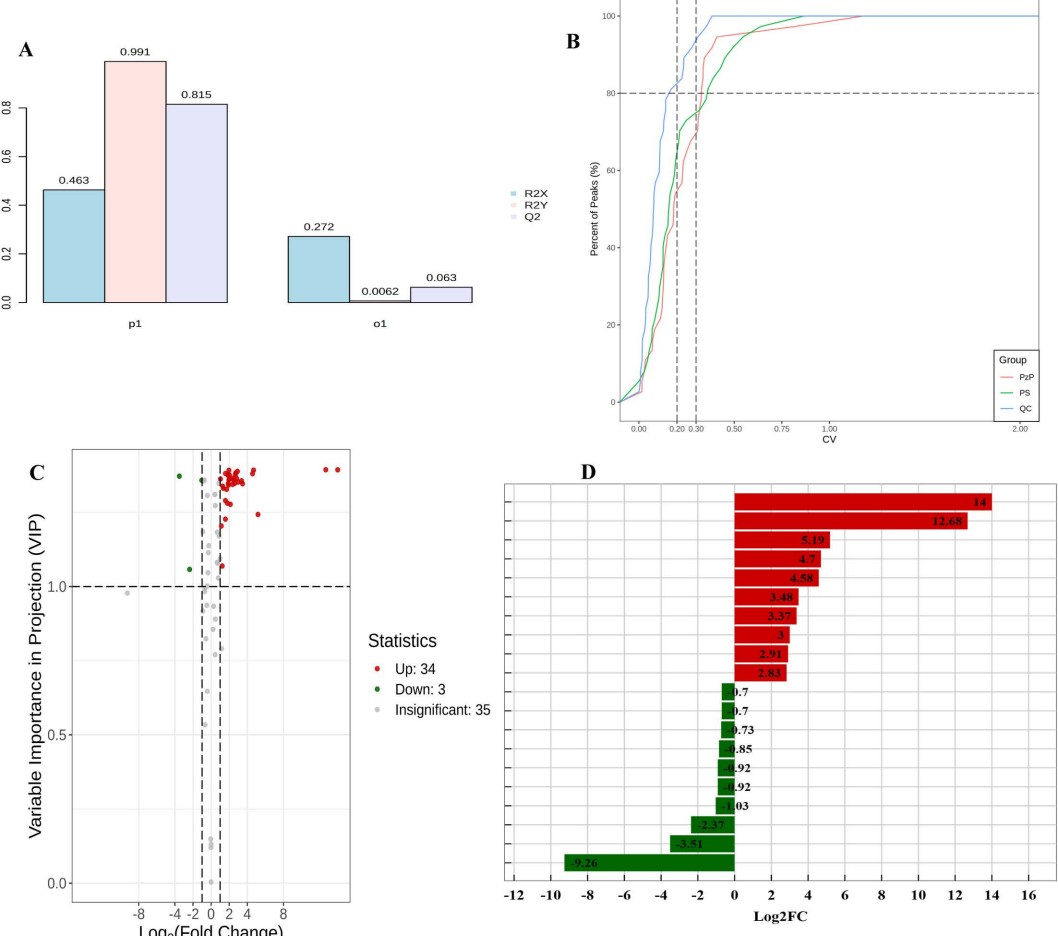

**Fig 2. Chemical differences between PzP and PS. A: Score plot of the Orthogonal Partial Least Squares Discriminant Analysis (OPLS-DA) model.** The model summary plot shows the R2X, R2Y, and Q2 corresponding to the predictive and orthogonal components in the OPLS-DA model. In the horizontal axis, P1 represents the predictive component, O1 represents the orthogonal component, and the vertical axis represents the corresponding R2X, R2Y, and Q2. B: Distribution plot of sample CVs. C: Volcano plot showing the differential expression levels of metabolites in PzP and PS samples. Red, green, and gray dots represent upregulated, downregulated, and non-significant differential expression of metabolites, respectively. D: The top 10 upregulated and downregulated compounds in PzP vs PS. Red and green colors represent upregulation and downregulation of metabolites, respectively.

their derivatives significantly accumulate in PzP. The significant differential metabolites in this study between PzP and PS were γ-glutamylmethionine and N-Acetyl-L-Glutamine (Fig 2D).

**The content of amino acids and their derivatives differs among various species of *Polygonatum*.** Our research focuses on examining the composition of amino acids and their derivatives in various species of *Polygonatum*. We employed a metabolomic analysis technique utilizing LC-MS/MS to detect and quantify 37 amino acids and their derivatives (Table 1, Fig 3). We found that in PzP, N-Acetyl-L-Glutamine, γ-glutamylmethionine, L-Arginine, N-Alpha-Acetyl-L-Asparagine, N-α-Acetyl-L-ornithine, N6-Acetyl-L-lysine, S-(Methyl)glutathione, 5-Oxoproline, DL-Methionine, L-Ornithine, L-Methionine, L-Glutamine, L-Lysine, N-Monomethyl-L-Arginine, Arginine methyl ester, Homoarginine, N-Acetyl-L-glutamic acid, Trimethyllysine, S-Methyl-L-cysteine, L-Histidine, L-Tryptophan, L-Leucine, L-Isoleucine, L-Tyrosine, L-Norleucine, L-Saccharopine, L-Threonine, L-Methionine methyl ester, NG,NG-Dimethyl-L-Arginine, N,N-Dimethylglycine, L-Phenylalanine, L-Glutamine-O-glycoside, N,N'-Dimethylarginine, 5-Hydroxy-L-tryptophanreached the highest values, while L-Valyl-L-Leucine, S-(5'-Adenosy)-L-homocysteine and N-acetyl-L-tyrosinereached the highest values in PS.

## Analysis of metabolic pathways for different amino acids and their derivatives

The KEGG database was utilized to perform enrichment and analysis of differential amino acids and their derivatives in various samples to obtain comprehensive functional information for PzP vs PS. There were a total of 38 metabolic pathways were enriched with differential amino acids and their derivatives. The top 20 metabolic pathways are significantly associated with key biosynthetic processes, such as Metabolic pathways, Pyrimidine metabolism, Aminoacyl-tRNA biosynthesis, Biosynthesis of amino acids, Glucosinolate biosynthesis, Valine, leucine and isoleucine biosynthesis, 2-Oxocarboxylic acid metabolism, Phenylalanine metabolism, Biosynthesis of secondary metabolites, Phenylpropanoid biosynthesis, Arginine biosynthesis, etc. (Fig 4). Further research on the differential metabolites of amino acids and their

**Table 1. Relative content of amino acids and their derivatives in PzP vs PS.**

| Compounds | Q1 (Da) | Q3 (Da) | Molecular weight (Da) | Compounds | Q1 (Da) | Q3 (Da) | Molecular weight (Da) |
|---|---|---|---|---|---|---|---|
| N-Acetyl-L-Glutamine | 187.07 | 125.00 | 188.0797 | L-Histidine | 156.08 | 110.00 | 155.0695 |
| γ-glutamylmethionine | 279.10 | 133.03 | 278.0936 | L-Tryptophan | 203.08 | 116.05 | 204.0899 |
| L-Arginine | 175.12 | 116.00 | 174.1117 | L-Leucine | 132.10 | 86.20 | 131.0946 |
| N-Alpha-Acetyl-L-Asparagine | 173.06 | 58.03 | 174.0641 | L-Isoleucine | 132.10 | 86.00 | 131.0946 |
| N-α-Acetyl-L-ornithine | 173.11 | 131.08 | 174.1004 | L-Tyrosine | 182.08 | 136.10 | 181.0739 |
| N6-Acetyl-L-lysine | 189.12 | 126.00 | 188.1161 | L-Norleucine | 132.10 | 86.00 | 131.0946 |
| S-(Methyl)glutathione | 322.11 | 130.00 | 321.0995 | L-Saccharopine | 275.00 | 257.00 | 276.1321 |
| 5-Oxoproline | 128.04 | 82.03 | 129.0426 | L-Threonine | 120.07 | 74.00 | 119.0582 |
| DL-Methionine | 150.10 | 61.00 | 149.0510 | L-Methionine methyl ester | 164.07 | 104.00 | 163.0667 |
| L-Ornithine | 133.00 | 116.00 | 132.0899 | NG,NG-Dimethyl-L-arginine | 203.15 | 70.07 | 202.1430 |
| L-Methionine | 150.06 | 61.00 | 149.0511 | N,N-Dimethylglycine | 104.07 | 86.00 | 103.0633 |
| L-Glutamine | 147.08 | 84.00 | 146.0691 | L-Phenylalanine | 166.09 | 120.08 | 165.0705 |
| L-Lysine | 147.11 | 84.00 | 146.1055 | L-Glutamine-O-glycoside | 307.00 | 145.00 | 308.1220 |
| N-Monomethyl-L-arginine | 189.13 | 70.07 | 188.1273 | N,N'-Dimethylarginine;SDMA | 203.15 | 70.07 | 202.1430 |
| Arginine methyl ester | 189.14 | 70.07 | 188.1273 | 5-Hydroxy-L-tryptophan | 221.09 | 204.00 | 220.0848 |
| Homoarginine | 189.13 | 144.00 | 188.1273 | L-Valyl-L-Leucine | 231.16 | 72.08 | 230.1630 |
| N-Acetyl-L-glutamic acid | 188.06 | 128.00 | 189.0637 | S-(5'-Adenosy)-L-homocysteine | 385.13 | 250.00 | 384.1216 |
| Trimethyllysine | 189.16 | 84.08 | 188.1525 | N-Acetyl-L-tyrosine | 224.09 | 136.00 | 223.0845 |
| S-Methyl-L-cysteine | 136.04 | 91.05 | 135.0354 | | | | |

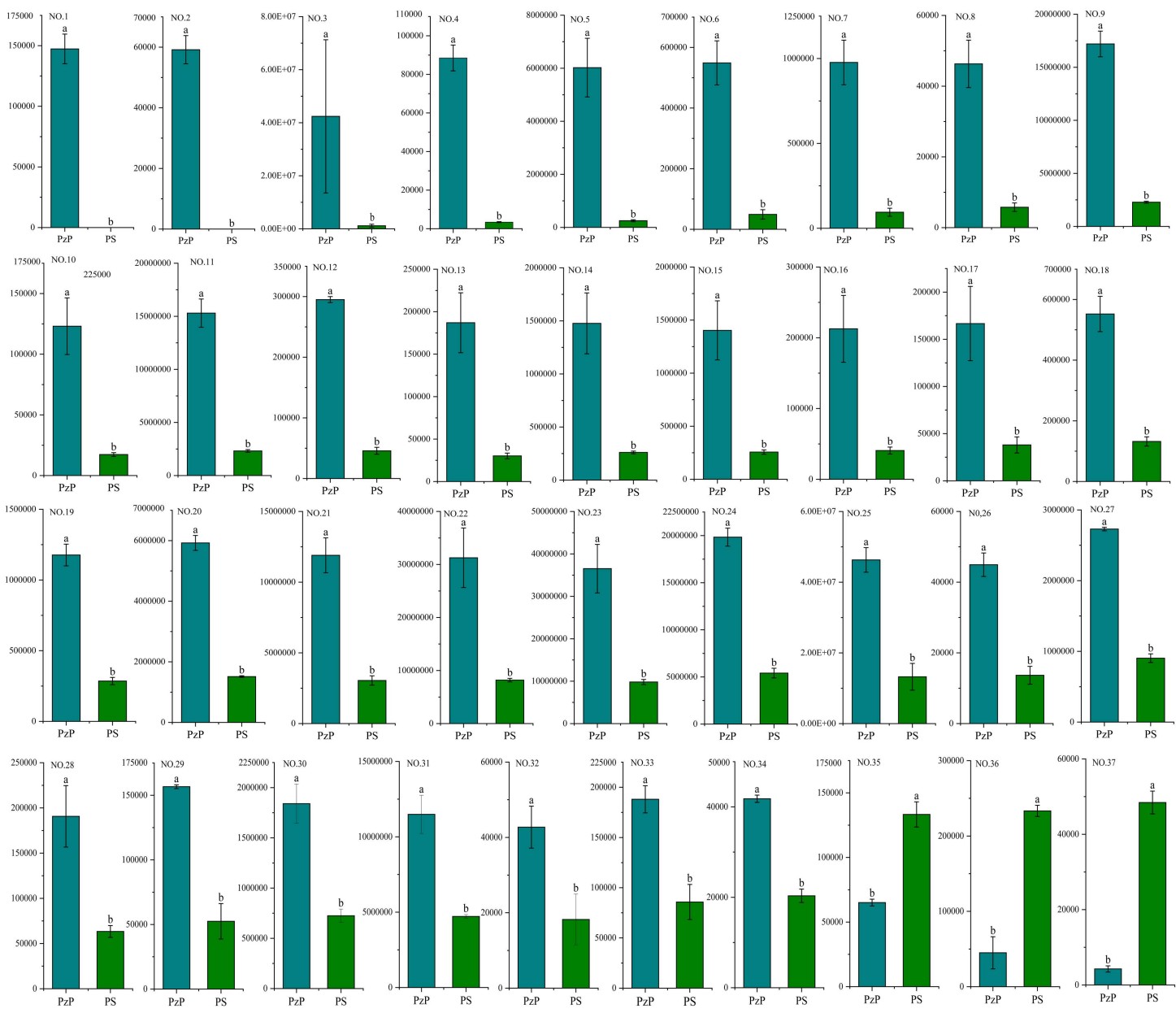

**Fig 3. Peak areas of the 37 amino acids and their derivatives metabolites identified in the PzP vs PS varieties.** Significance was assessed using Duncan's test. Different letters indicate significant differences (P < 0.05).

derivatives through the KEGG database elucidates the metabolic levels of different biological samples and the significant changes in metabolites.

## Analysis of gene expression in the metabolic pathways of amino acids and their derivatives in *Polygonatum*

The carbon skeletons of amino acids and their derivatives are synthesized from glucose through the tricarboxylic acid (TCA) cycle and the pentose phosphate pathway (PPP). Based on the reported biosynthetic pathways of amino acids and their derivatives in other species, as well as the transcriptomic and metabolomic data of PzP vs PS, we analyzed the

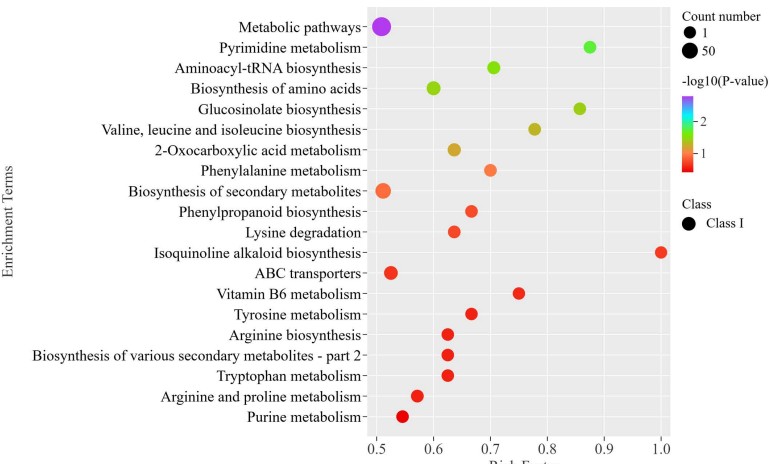

**Fig 4. Metabolic pathway analysis of differential metabolites of amino acids and their derivatives.**

biosynthetic pathways of amino acids and their derivatives in PzP vs PS (Fig 5). A total of 271 unigenes (S4 Table) encoding 47 enzymes from the transcriptome and 37 metabolites from the metabolome involved in the synthesis of amino acids and their derivatives were identified.

### Weighted Gene Co-expression Network Analysis (WGCNA) and candidate gene screening

In this study, we utilized the WGCNA clustering tool to cluster genes with similar expression patterns. We clustered 2539 genes detected in the 37 metabolic pathways where amino acids and their derivatives are located (S5 Table). We performed WGCNA analysis on the transcriptome of PzP vs PS, dividing differentially expressed genes into six modules (Fig 6A–6B). Genes enriched in each module have similar expression patterns and are represented by a specific color. Furthermore, we conducted correlation analysis between different modules and traits to explore gene clusters related to amino acid synthesis. The module with the highest number of enriched genes is MEturquoise, with 1178 genes, while the MEgrey module has the fewest enriched genes, with 50 genes. The module-trait relationship analysis results (Fig 6C) show that MEblue is highly positively correlated with 34 amino acids and their derivatives (p < 0.05, the same below), with correlations ranging from 0.86 to 1, and MEturquoise is highly positively correlated with L-Valyl-L-Leucine, S-(5'-Adenosy)-L-homocysteine, and N-Acetyl-L-tyrosine, with correlations of 0.97, 0.95, and 0.98, respectively. Using Cytoscape software, we networked the interrelationships between modules and selected candidate genes based on their connectivity within the network. Among them, Cluster-39698.1, Cluster-27133.0, Cluster-28145.0, Cluster-51791.5, luster-29705.11, and Cluster-53239.2 in the MEblue module are candidate genes; Cluster-49006.0, Cluster-35441.2, Cluster-17104.0, and Cluster-49223.2 in the MEturquoise module are candidate genes (Fig 6D–6E, S6 Table).

### Joint analysis of transcriptal and metabolic groups

Fig 7A depicts the top 20 genes that show significant differences and the corresponding network maps of all metabolites involved in the synthesis pathways of amino acids and their derivatives.L-Valyl-L-Leucine and N-Acetyl-L-tyrosine exhibited negative correlations, but other amino acids and derivatives displayed stronger positive correlations and statistically significant connections. Fig 5B displays a network diagram illustrating 10 crucial genes and differential metabolites that are involved in the production of amino acids and their derivatives. Each of these genes and metabolites is firmly linked to the corresponding amino acid and derivative. Fig 5C displays a heat map that highlights the enrichment of 10 important genes and differential metabolites related to the production of amino acids and their derivatives.

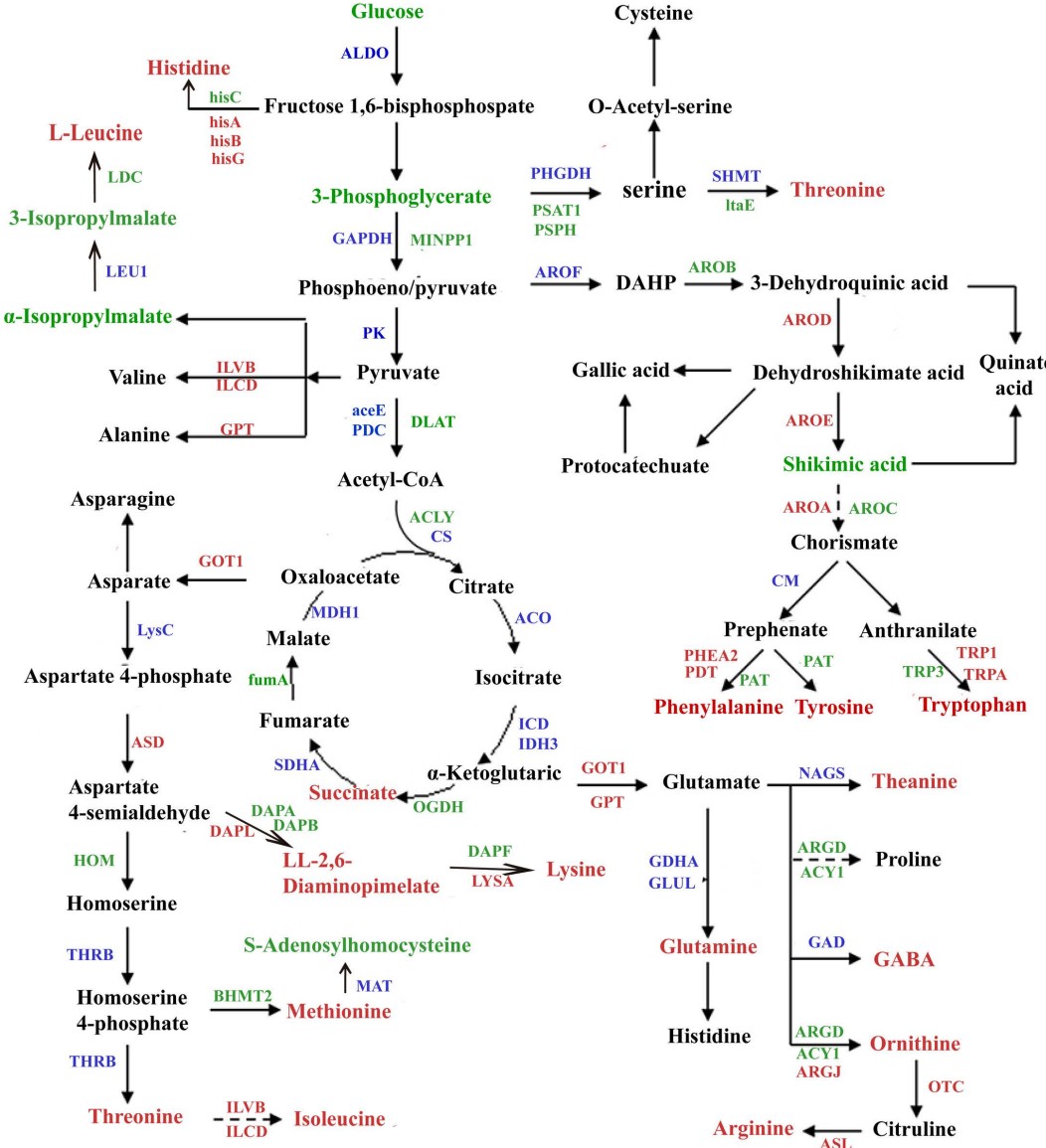

**Fig 5. The relationship between key genes in the biosynthetic pathways of amino acids and their derivatives and the synthesis of metabolites.**
Red font indicates upregulation in PzP, green font indicates downregulation, and blue indicates detected but with little change.

## Validation of DEGs by qRT–PCR

To verify the accuracy of the transcriptome data, we conducted RNA sequencing analysis and reverse transcription PCR (qRT-PCR) on the genes related to the synthesis of amino acids and their derivatives, selected from PzP and PS, respectively, to determine the authenticity and reliability of the transcriptome data. We selected 10 key genes for qRT-PCR validation. Despite some minor variations between the RNA-Seq and qRT-PCR results, the overall pattern was consistent for roughly 80% of the genes (Fig 8). Therefore, transcriptome sequencing is considered reliable.

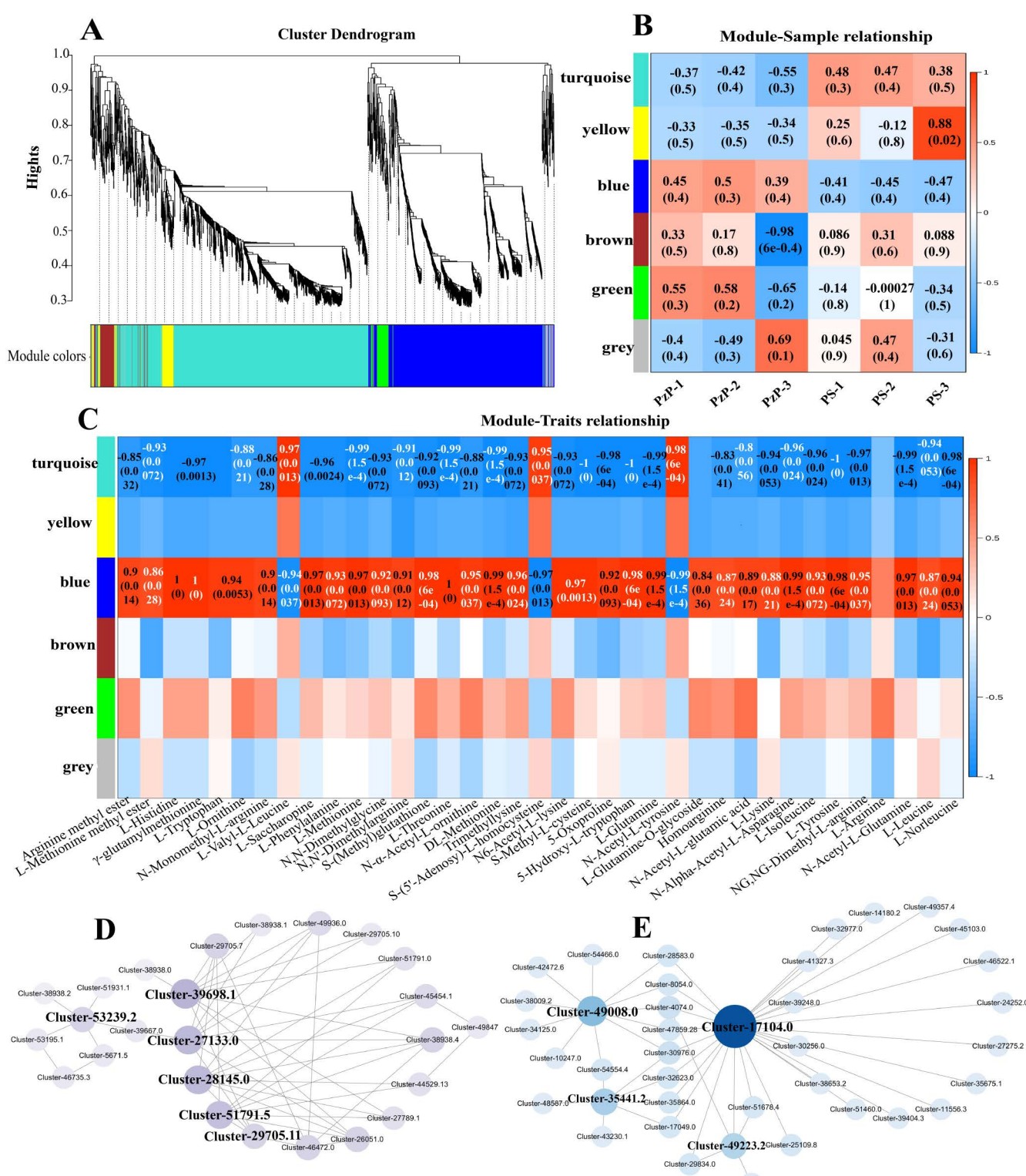

**Fig 6. Co-expression analysis of transcripts and active components in all pathways of amino acids and their derivatives.** A: Hierarchical cluster map of co-expression modules. B: Heatmap of sample and module correlations. C: Correlation heatmap between amino acids, their derivatives, and modules. D: Network graph of the top 50 gene connections in the MEblue module. E: Network graph of the top 50 gene connections in the MEturquoise module.

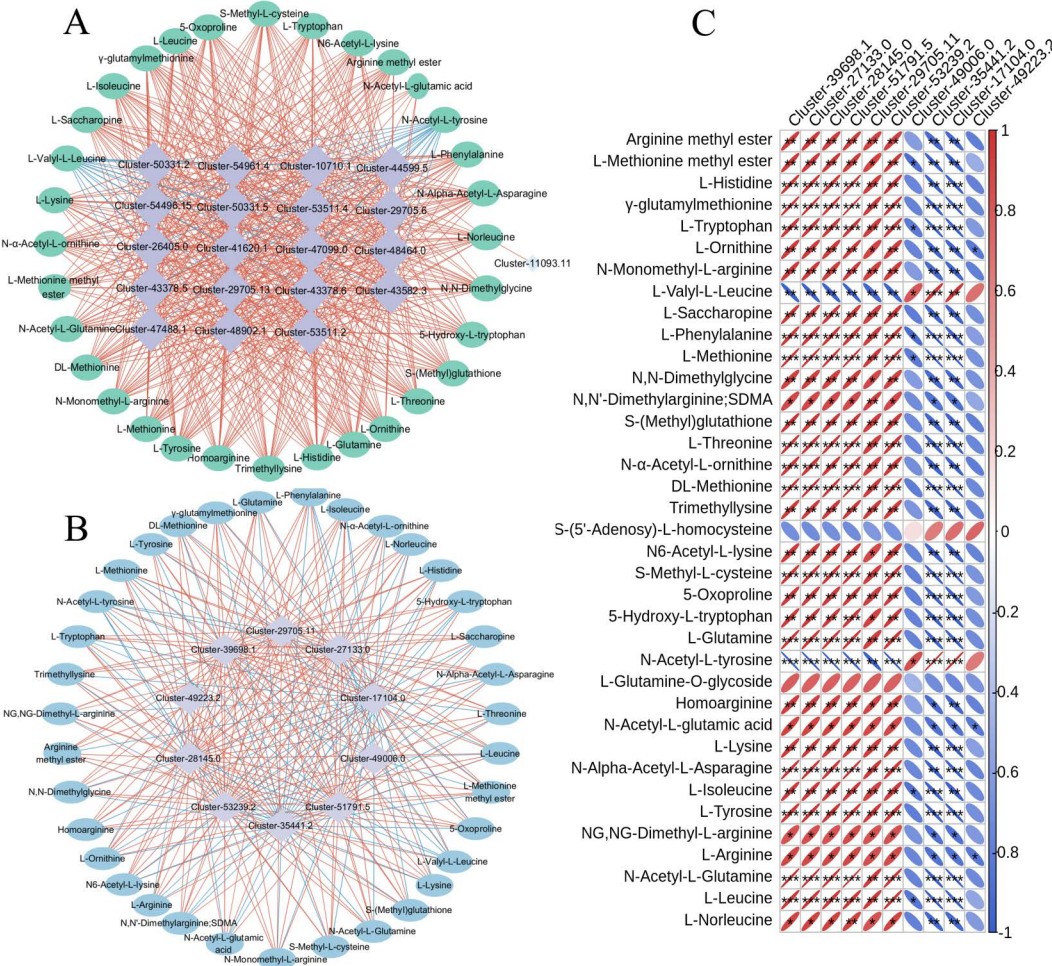

**Fig 7. Correlation analysis.** A: The top 20 differential genes associated with amino acid and derivative synthesis pathways and the related network of all differentials metabolites.B: 10 key genes related to amino acid and derivative synthesis pathways are related to all differential metabolites.C: Related hot maps of 10 key genes and differential metabolites involved in the synthesis of amino acids and derivatives.

## Discussion

### Analysis of amino acids and their derivative metabolites

Current research on amino acids in *Polygonatum* species has primarily focused on their composition and nutritional content, with limited insight into the molecular mechanisms underlying their biosynthesis and regulation [16]. As *Polygonatum* rhizomes develop underground, photosynthetically derived assimilates from aerial parts must be transported to the rhizomes to support their growth and storage functions [17]. These assimilates include carbohydrates, lipids, proteins, and notably, free amino acids and their derivatives, all of which contribute significantly to rhizome development and metabolic homeostasis [18,19]. Studies have found that there are 18 amino acids in *Polygonatum sibiricum* [20], and 17 amino acids were detected in wild *Polygonatum sibiricum* [21], which have a high edible value. However, systematic metabolomic profiling of amino acids and their derivatives across different *Polygonatum* species remains scarce. With the advent of advanced LC-MS/MS techniques, broad-spectrum targeted metabolomics has become a powerful approach for dissecting complex metabolic networks in plants [22,23]. This study utilized LC-MS/MS-based broad-targeted metabolomics to profile

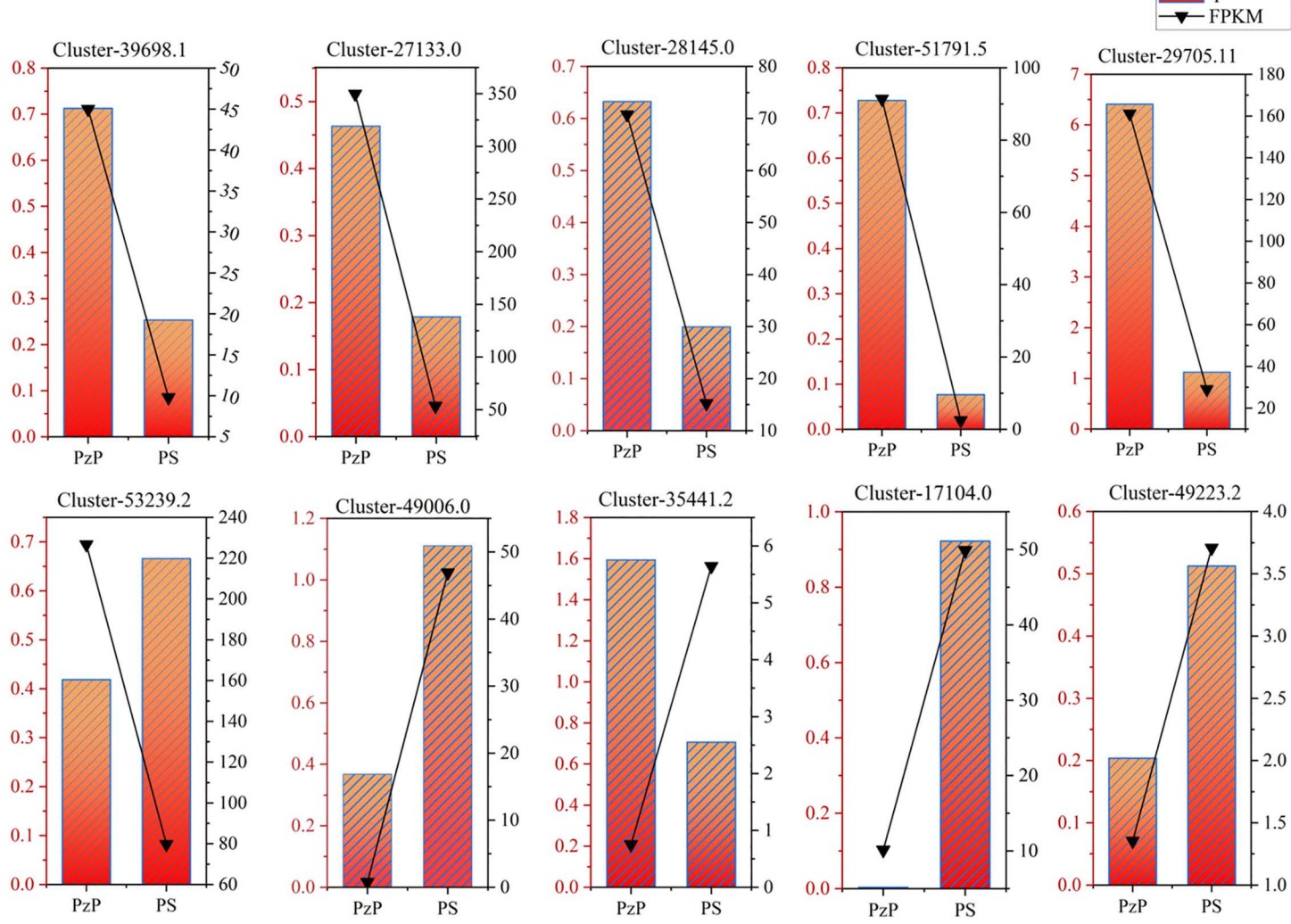

**Fig 8. Results of qRT-PCR assay.** The line graph represents the FPKM results of transcriptome data; the bar graph represents the qRT-PCR results.

amino acid-related metabolites in *Polygonatum* species. Seventy two amino acids and their derivatives were identified; 37 exhibited significant differential accumulation between species. Strikingly, 34 were found at higher levels in PzP, while only 3 showed higher accumulation in PS. This striking contrast suggests species-specific differences in amino acid biosynthesis and regulation, providing a foundation for uncovering the underlying genetic and biochemical mechanisms.

## Biosynthetic pathways and physiological roles of amino acids

Previous studies have revealed that *Polygonatum* species exhibit divergent amino acid profiles [7,24], which may be linked to their specific physiological roles. For example, arginine plays a crucial role in nitrogen (N) storage and transport; during rhizome development, the catabolism of arginine is essential for the release of nitrogen from source tissues and induces the expression of a suite of related enzymes [25,26]. Glutamate serves as a core precursor for a wide array of metabolic compounds, including proteinogenic amino acids (e.g., proline, arginine, histidine), non-protein amino acids (e.g., γ-aminobutyric acid, GABA), antioxidant tripeptides (e.g., glutathione), heme, chlorophyll, and others [27,28]. Tryptophan is a critical

precursor for numerous bioactive compounds such as auxins, tryptophan derivatives, phytoanticipins, terpenoids, and alkaloids, and is thus integral to plant growth, development, and stress responses [29–31]. Our metabolomic data revealed that PzP exhibited elevated levels of several amino acids and derivatives, particularly glutamate, arginine, ornithine, and tryptophan, compared to PS. These amino acids function as protein building blocks and act as metabolic signals or precursors to diverse secondary metabolites, potentially contributing to the enhanced growth performance, stress resilience, and medicinal value of PzP. For instance, higher glutamate and arginine levels may promote more efficient nitrogen cycling and storage [32]. Ornithine may enhance polyamine biosynthesis, which supports cell division and rhizome expansion [33]. Moreover, tryptophan is a well-established indole-3-acetic acid (IAA) precursor, and its elevated content may promote auxin biosynthesis, thereby influencing root development and hormonal signaling [34]. Furthermore, these amino acids may regulate the biosynthesis of additional compound classes, implying a broader impact on metabolic homeostasis and adaptive traits. To explore potential regulatory mechanisms, transcriptomic integration identified 271 genes encoding 47 enzymes involved in amino acid biosynthesis. The higher accumulation of metabolites in PzP likely stems from enhanced expression of these biosynthetic genes. Our result indicates a more active transcriptional landscape in PzP, potentially modulated by upstream signals related to nitrogen metabolism, hormonal pathways, or environmental stress responses.

### Screening of target genes involved in the synthesis of amino acids and their derivatives

Transcription factors (TFs) play a pivotal role in regulating genes involved in amino acid metabolism, and are also crucial for rhizome development, cellular differentiation, and environmental adaptation. Previous studies in *Arabidopsis* and rice have shown that specific TFs are tightly associated with various growth stages and responses to environmental stimuli [35]. Weighted Gene Co-expression Network Analysis (WGCNA) is an algorithm that discovers genes with similar expression patterns and classifies them into modules based on their distinct expression patterns [36]; genes with similar expression patterns are controlled by the same or similar factors, their functions are closely related, and they are even members of the same signal pathway or biological process, thus jointly exerting specific biological functions [37]. The transcriptomic data revealed six distinct gene modules in *Polygonatum*, among which the MEturquoise and MEblue modules exhibited strong positive correlations with key amino acid metabolites and their derivatives. Within these modules, highly connected (hub-like) genes were identified: six genes in MEblue (e.g., Cluster-39698.1, Cluster-27133.0) and four in MEturquoise (e.g., Cluster-49006.0, Cluster-35441.2). These hub genes are considered key regulators due to their central position in the co-expression network and potential role in modulating amino acid biosynthetic pathways [38,39]. The elevated expression of these hub genes in PzP suggests a transcriptional basis for its higher amino acid content. These genes may encode rate-limiting enzymes or regulatory proteins (e.g., TFs, transporters, or cofactors) that promote flux through amino acid biosynthetic pathways. These findings provide a functional basis for the observed metabolic divergence and suggest that these genes represent valuable targets for genetic improvement. In conclusion, the differential expression of amino acid-related genes between PzP and PS reveals potential mechanisms driving metabolic differences and offers a genetic foundation for germplasm selection, metabolic engineering, and trait improvement in *Polygonatum* species.

## Results

Metabolomic analysis of PzP vs PS identified 72 amino acids and derivatives, with 37 showing significant differences; 34 were highest in PzP and 3 in PS. From amino acid synthesis pathways, 272 genes encoding 47 related enzymes were identified. WGCNA of 2539 unigenes from 37 KEGG pathways revealed six modules, with amino acid components positively correlated with the MEturquoise and MEblue modules. Key genes such as Cluster-39698.1, Cluster-27133.0, and Cluster-49006.0 showed high connectivity. Ultimately, 10 genes were identified as most likely to affect the synthesis of amino acids and their derivatives. After bioinformatics analysis, qRT-PCR was conducted to confirm the results. It may be possible to use these genes in the future as candidate genes for genetic and metabolic engineering research in order to increase biosynthesis of amino acids and their derivatives in *Polygonatum*.

## Supporting information

**S1 Table. qRT-PCR primer list.**
(XLSX)

**S2 Table. Total amino acids and their derivatives.**
(XLSX)

**S3 Table. Differential amino acids and their derivatives.**
(XLSX)

**S4 Table. Differential genes in the biosynthetic pathways of amino acids and their derivatives.**
(XLSX)

**S5 Table. Total genes in the biosynthetic pathways of amino acids and their derivatives.**
(XLSX)

**S6 Table. The CDS sequence of the target gene.**
(XLSX)

## Acknowledgments

We are deeply in debted to Professor Lingjun Cui for helpful suggestions and comments on bioinformatic analyses, and we thank Professor Qing Xiao for valuable comments on previous versions of the manuscript.

## Author contributions

**Conceptualization:** Xiuzhi Wang.

**Data curation:** Qiang Xiao.

**Formal analysis:** Xiuzhi Wang.

**Funding acquisition:** Lingjun Cui, Qiang Xiao.

**Investigation:** Xiuzhi Wang.

**Methodology:** Xiuzhi Wang.

**Project administration:** Qiang Xiao.

**Resources:** Qiang Xiao.

**Supervision:** Lingjun Cui.

**Validation:** Xiuzhi Wang.

**Writing – original draft:** Xiuzhi Wang.

**Writing – review & editing:** Qiang Xiao.

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
