## [Decision Letter · Decision Letter 0]

Dear Dr. Xiao,

Thank you for submitting your manuscript to PLOS ONE. After careful consideration, we feel that it has merit but does not fully meet PLOS ONE’s publication criteria as it currently stands. Therefore, we invite you to submit a revised version of the manuscript that addresses the points raised during the review process.

We look forward to receiving your revised manuscript.

Kind regards,

Vikas Sharma, Ph.D

Academic Editor

PLOS ONE

**Journal Requirements:**

the National Natural Science Foundation of China (31260057), Natural Science Foundation of Hubei Province (Joint Fund) (2023AFD077), The Open Fund of Hubei Key Laboratory of Biological Resources Protection and Utilization (Hubei Minzu University) (KYPT012502).

3. In the online submission form, you indicated that "We deeply acknowledge the fundamental importance of data sharing in scientific research. However, due to our inadequate preservation of the original data, they have passed their validity period, and as a result, we are unable to provide a transcription of the original data, for which we sincerely apologize. Should you need any other data, please do not hesitate to contact our corresponding author at any time to request it."

**Additional Editor Comments:**

Please check the format of manuscript and make it according to journal's guidelines.

Please divide discussion into subheadings.

Reviewers' comments:

Reviewer's Responses to Questions

1. Is the manuscript technically sound, and do the data support the conclusions?

Reviewer #1: Yes

2. Has the statistical analysis been performed appropriately and rigorously?

Reviewer #1: Yes

3. Have the authors made all data underlying the findings in their manuscript fully available?

Reviewer #1: Yes

4. Is the manuscript presented in an intelligible fashion and written in standard English?

Reviewer #1: Yes

**Reviewer #1: ** The submitted MS explains the thorough study of the metabolome and transcriptome analysis on two species of Polygonatum. The overall manuscript has been written very well and the designed experiments are scientifically sound. The major concern is with two points:

1. All the pictures are blurred, hence, i can not correlate the findings written in ms to the figures. Therefore, authors are advised to kindly provide the high resolution figures wherever it is possible because in large dataset, some picture can not be so visible that i understand.

2. All the provided deliverables are generated bioinformatically which is not so reliable till it will not be validated by wet lab experiments. The study could be validated at least targeting few amino acids gene expression analysis.

After the revised figures submission, i would like to review the MS to reach any decision.

Do you want your identity to be public for this peer review? For information about this choice, including consent withdrawal, please see our Privacy Policy

Reviewer #1: Yes: Deepanshu Jayaswal

---

## [Author Response · Author response to Decision Letter 1]

5 Feb 2025

Dear Editor,

Thank you very much for kindly providing us with the opportunity to revise our manuscript entitled "Integrative analysis of the metabolome and transcriptome provides insights into the mechanisms of amino acids and their derivatives biosynthesis in Polygonatum". The reviewers’ comments are all valuable and very helpful for revising and improving our paper. The revised sections are highlighted in the revision. The main corrections in the paper and the response to the editor and reviewer’s comments are as follows:

Response to Editor and Reviewers:

Editor:

1.Please check the format of manuscript and make it according to journal's guidelines.

Response: We sincerely thank Editor for the meticulous comment. We have reviewed the manuscript carefully and ensure that its format aligns with the journal's guidelines. Please let me know if there are any specific areas you would like me to pay extra attention to.

2.Please divide discussion into subheadings.

Response: We sincerely thank Editor for the meticulous comment. Following your valuable suggestion, we have carefully divided the discussion section into subheadings as requested. Please find the revised conclusion in lines 321-390 of the updated manuscript.

Reviewer #1:

1. All the pictures are blurred, hence, i can not correlate the findings written in ms to the figures. Therefore, authors are advised to kindly provide the high resolution figures wherever it is possible because in large dataset, some picture can not be so visible that i understand.

Response: We sincerely thank you for the meticulous comment. We have taken the necessary steps to enhance the clarity , quality and font size of the figures. Thank you for your valuable comments.

2. All the provided deliverables are generated bioinformatically which is not so reliable till it will not be validated by wet lab experiments. The study could be validated at least targeting few amino acids gene expression analysis.

Response: We sincerely thank you for the meticulous comment. In response to your valuable suggestions, we have already utilized qPCR technology to conduct expression analysis on the 10 target genes selected in the manuscript. Through experimental validation, our study has become more comprehensive and reliable. Please find the revised conclusion in lines 138-155 and 309-319 of the updated manuscript.

Other revisions to the article: Furthermore, after careful consideration of your valuable review comments, we have not only made revisions in accordance with your suggestions but also further enhanced the results to provide a more concise summary of the article. We believe these additional modifications contribute to improving the quality of the research and the clarity of the manuscript. For further details, please refer to lines 392-402 of the revised manuscript.

In addition, the complete statement of the funding disclosure is as follows: “ this study was jointly funded by the National Natural Science Foundation of China [NSFC, grant number 31260057, QX), Natural Science Foundation of Hubei Province (Joint Fund) [NSFHP, grant number 2023AFD077, QX], The Open Fund of Hubei Key Laboratory of Biological Resources Protection and Utilization (Hubei Minzu University) [OHBP, grant number KYPT012403, LJC]. The funders had no role in study design, data collection and analysis, decision to publish, or preparation of the manuscript.” The role of the funders is stated in the last sentence.

---

## [Decision Letter · Decision Letter 1]

Dear Dr. Xiao,

Thank you for submitting your manuscript to PLOS ONE. After careful consideration, we feel that it has merit but does not fully meet PLOS ONE’s publication criteria as it currently stands. Therefore, we invite you to submit a revised version of the manuscript that addresses the points raised during the review process.

We look forward to receiving your revised manuscript.

Kind regards,

Mojtaba Kordrostami, Ph.D.

Academic Editor

PLOS ONE

Journal Requirements:

Reviewers' comments:

Reviewer's Responses to Questions

**Comments to the Author**

Reviewer #1: All comments have been addressed

Reviewer #2: All comments have been addressed

2. Is the manuscript technically sound, and do the data support the conclusions?

Reviewer #1: Yes

Reviewer #2: Yes

3. Has the statistical analysis been performed appropriately and rigorously?

Reviewer #1: Yes

Reviewer #2: Yes

4. Have the authors made all data underlying the findings in their manuscript fully available?

Reviewer #1: Yes

Reviewer #2: Yes

5. Is the manuscript presented in an intelligible fashion and written in standard English?

Reviewer #1: Yes

Reviewer #2: Yes

Reviewer #1: The authors have addressed the raised comments which are satisfactory. Therefore, the MS in the current form may be accepted.

Reviewer #2: The authors were able to accurately identify the research gap in the field of amino acid metabolites in Polygonatum species, and a significant number of genes related to amino acid biosynthesis pathways were reported, indicating extensive data mining and in-depth analysis, But there are a few points that would make the manuscript richer if addressed.

The results are presented descriptively and rarely involve mechanistic or functional interpretation.

Why are metabolite levels higher in PzP? Is it related to genetic differences, developmental conditions, or expression of regulators?

Can changing tryptophan levels affect levels of hormones such as IAA?

The discussion should end with a more practical conclusion. For example: could this difference in metabolites be the basis for germplasm selection or genetic modification?

**Do you want your identity to be public for this peer review?** For information about this choice, including consent withdrawal, please see our Privacy Policy

Reviewer #1: **Yes: ** Deepanshu Jayaswal

Reviewer #2: No

---

## [Author Response · Author response to Decision Letter 2]

15 May 2025

Response to Editor and Reviewers:

Editor:

1.Please review your reference list to ensure that it is complete and correct. If you have cited papers that have been retracted, please include the rationale for doing so in the manuscript text, or remove these references and replace them with relevant current references. Any changes to the reference list should be mentioned in the rebuttal letter that accompanies your revised manuscript. If you need to cite a retracted article, indicate the article’s retracted status in the References list and also include a citation and full reference for the retraction notice.

Response: We sincerely thank Editor for the reminder regarding the accuracy and completeness of the reference list. We have carefully reviewed all references cited in the manuscript. None of the references have been retracted. For references originally sourced from CNKI ([2], [8], [21]), we have ensured that they are cited with complete bibliographic details. These references were retained because they provide critical local or species-specific data that are not available in English literature. All necessary updates have been made in the revised reference list.

Reviewer #2:

1. The results are presented descriptively and rarely involve mechanistic or functional interpretation.

Response: We sincerely thank Reviewer #2 for the valuable suggestion. we have added mechanistic and functional interpretations of the key findings to enrich the discussion. These revisions can be found in the “Discussion” section, particularly under “Biosynthetic Pathways and Physiological Roles of Amino Acids.” We hope this has strengthened the biological relevance and interpretative depth of our results.

2. Why are metabolite levels higher in PzP? Is it related to genetic differences, developmental conditions, or expression of regulators?

Response: We sincerely thank Reviewer #1 for the insightful question. IIn the revised manuscript, we have addressed this point in the Discussion section. We propose that the elevated metabolite levels observed in PzP may be primarily attributed to transcriptional regulation. Integration of transcriptomic and metabolomic data revealed that key biosynthetic genes were more highly expressed in PzP. In addition, co-expression network analysis identified several hub genes potentially involved in regulatory control of amino acid metabolism, which were also upregulated in PzP. These findings suggest that the metabolic differences are likely due to genetic and regulatory factors, rather than only environmental or developmental conditions.

3. Can changing tryptophan levels affect levels of hormones such as IAA?

Response: We sincerely thank Reviewer #2 for the important point. We agree that tryptophan is a key precursor in the indole-3-acetic acid (IAA) biosynthesis pathway in plants. In the revised discussion section, we have highlighted this relationship and proposed that the elevated tryptophan levels observed in PzP may potentially lead to increased auxin biosynthesis, which could influence rhizome development and stress responses. Although we did not directly measure IAA in this study, this connection warrants future investigation. Please find the revised conclusion in lines 360-370 of the updated manuscript.

4. The discussion should end with a more practical conclusion. For example: could this difference in metabolites be the basis for germplasm selection or genetic modification?

Response: We sincerely thank Reviewer #2 for the insightful suggestion. We have revised the final paragraph of the discussion to include a more practical conclusion, emphasizing the potential application of our findings in germplasm selection and genetic improvement. In particular, we highlight that the identified key amino acid-related metabolites and biosynthetic genes may serve as valuable targets for future breeding or metabolic engineering efforts aimed at enhancing the nutritional and medicinal qualities of Polygonatum. The revised discussion now ends with a clear statement of these implications.

---

## [Decision Letter · Decision Letter 2]

Integrative analysis of the metabolome and transcriptome provides insights into the mechanisms of amino acids and their derivatives biosynthesis in Polygonatum

PONE-D-24-36620R2

Dear Dr. Xiao,

We’re pleased to inform you that your manuscript has been judged scientifically suitable for publication and will be formally accepted for publication once it meets all outstanding technical requirements.

Kind regards,

Mojtaba Kordrostami, Ph.D.

Academic Editor

PLOS ONE

Additional Editor Comments (optional):

Reviewers' comments:

Reviewer's Responses to Questions

**Comments to the Author**

Reviewer #1: (No Response)

Reviewer #2: All comments have been addressed

2. Is the manuscript technically sound, and do the data support the conclusions?

Reviewer #1: Yes

Reviewer #2: Yes

3. Has the statistical analysis been performed appropriately and rigorously?

Reviewer #1: Yes

Reviewer #2: Yes

4. Have the authors made all data underlying the findings in their manuscript fully available?

Reviewer #1: Yes

Reviewer #2: Yes

5. Is the manuscript presented in an intelligible fashion and written in standard English?

Reviewer #1: Yes

Reviewer #2: Yes

Reviewer #1: Authors addressed the raised concerns sincerely. therefore, the MS may be accepted in the current form.

Reviewer #2: The authors have carefully incorporated the comments and suggestions of the reviewers. The revised version of the article is acceptable for publication in the journal.

**Do you want your identity to be public for this peer review?** For information about this choice, including consent withdrawal, please see our Privacy Policy

Reviewer #1: **Yes: ** Deepanshu Jayaswal

Reviewer #2: No

---

## [Editor Report · Acceptance letter]

PONE-D-24-36620R2

PLOS ONE

Dear Dr. Xiao,

I'm pleased to inform you that your manuscript has been deemed suitable for publication in PLOS ONE. Congratulations! Your manuscript is now being handed over to our production team.

Kind regards,

on behalf of

Dr. Mojtaba Kordrostami

Academic Editor

PLOS ONE